# Microsatellite Characteristics of Silver Carp (*Hypophthalmichthys*
*molitrix*) Genome and Genetic Diversity Analysis in Four Cultured Populations

**DOI:** 10.3390/genes13071267

**Published:** 2022-07-16

**Authors:** Yajun Wang, Hang Sha, Xiaohui Li, Tong Zhou, Xiangzhong Luo, Guiwei Zou, Yi Chai, Hongwei Liang

**Affiliations:** 1Engineering Research Center of Ecology and Agricultural Use of Wetland, Ministry of Education, College of Agriculture, College of Animal Science, Yangtze University, Jingzhou 434025, China; wangyajun@yfi.ac.cn; 2Yangtze River Fisheries Research Institute, Chinese Academy of Fishery Sciences, Wuhan 430223, China; sh1812@yfi.ac.cn (H.S.); lixiaohui@yfi.ac.cn (X.L.); zhoutong@yfi.ac.cn (T.Z.); lxz@yfi.ac.cn (X.L.); zougw@yfi.ac.cn (G.Z.)

**Keywords:** genome-wide identification, *Hypophthalmichthys molitrix*, SSR, genetic diversity, population structure

## Abstract

*Hypophthalmichthys molitrix* is one of the four most important fish in China and has high breeding potential. However, simple sequence repeat (SSR) markers developed on *H. molitrix* genome level for genetic diversity analysis are limited. In this study, the distribution characteristics of SSRs in the assembled *H. molitrix* genome were analyzed, and new markers were developed to preliminarily evaluate the genetic diversity of the four breeding populations. A total of 368,572 SSRs were identified from the *H. molitrix* genome. The total length of SSRs was 6,492,076 bp, accounting for 0.77% of the total length of the genome sequence. The total frequency and total density were 437.73 loci/Mb and 7713.16 bp/Mb, respectively. Among the 2–6 different nucleotide repeat types, SSRs were dominated by di-nucleotide repeats (204,873, 55.59%), and AC/GT was the most abundant motif. The number of SSRs on each chromosome was positively correlated with the length. The 13 pairs of markers developed were used to analyze the genetic diversity of four cultivated populations in Hubei Province. The results showed that the genetic diversity of the four populations was low, and the ranges of alleles (Na), effective alleles (Ne), observed heterozygosity (Ho), and Shannon’s index information (I) were 3.538–4.462, 2.045–2.461, 0.392–0.450, and 0.879–0.954, respectively. Genetic variation occurs mainly among individuals within populations (95.35%). UPGMA tree and Bayesian analysis showed that four populations could be divided into two different branches. Therefore, the genome-wide SSRs were effectively in genetic diversity analysis on *H. molitrix.*

## 1. Introduction

Silver carp (*H. molitrix*), as one of the four dominant fish in China, are mainly fed on phytoplankton and live in the upper and middle layers of water. They are widely distributed in ponds, lakes, rivers, and other major freshwater ecosystems in China and Asia [1]. Due to its fast growth, low breeding cost, high economic benefit, and purification role of water quality, it has become an important freshwater economic fish [2]. Before the implementation of the ten-year fishing ban in the Yangtze River, more attention was paid to the analysis of genetic diversity and genetic structure of natural populations, and less attention was paid to the genetic background of breeding populations. Moreover, breeding populations might affect the genetic diversity and adaptability of natural populations through the hatchery release of *H. molitrix* [3]. Therefore, it is necessary to further develop effective genetic markers to evaluate the genetic diversity of *H. molitrix* breeding populations. 

Microsatellites, also known as simple sequence repeats (SSRs), are DNA sequences composed of 1–6 bases as repeat units, which widely exist in the coding and non-coding regions of eukaryotic and prokaryotic genomes [4,5]. SSRs have been widely used in genetic relationship identification [6], population genetic structure analysis [7], genetic breeding [8], molecular-marker-assisted selection [9], gender marker detection [10], genetic linkage map construction [11], and other studies because of their high polymorphism, codominant inheritance, and easy detection. Traditional SSR markers are mainly developed by enrichment methods, genomic–SSR hybrid screening, and primer sharing of related species, but these methods are complicated in operation, long experimental cycle, and are easily affected by some human factors [12]. In recent years, with the rapid development of sequencing technology, a lot of the genomic information of non-model organisms has been reported, which provides basic data for the mass development of SSR markers [13]. 

So far, limited previous studies focused on SSRs markers development of *H. molitrix*. One hundred and fifty-nine SSR sites were obtained by transcriptome sequences of *H.*
*molitrix* [14]. Guo et al. [15] developed 134 polymorphic SSR markers and used 40 pairs for population genetic diversity analysis. However, the screening and development of SSR markers at the *H. molitrix* genome level have not been studied. In the present study, we screened SSR repeats at the whole *H. molitrix* genome level (unpublished, in our lab), analyzed the number, frequency, and type of SSRs, developed 13 SSR markers, and analyzed the genetic diversity of four *H. molitrix* cultivated populations. The study provides SSR characteristics of the *H. molitrix* genome and useful markers for analysis in genetic diversity and germplasm resources protection and utilization.

## 2. Materials and Methods

### 2.1. Sample Collections and DNA Extraction 

One hundred and twenty samples of four cultured *H. molitrix* populations were collected from Shishou (SS), Wuhan (WH), Xiaochang (XC), and Yaowan (YW) (Table 1), respectively, in Hubei Province in 2021. The tail fins were sampled and stored in anhydrous ethanol at −20 °C. Genomic DNA of samples were extracted using a high-salt method [16]. After extraction, the quality of the DNA was detected by 1% agarose gel electrophoresis and a UV gel imaging system. DNA concentrations were measured by NanoPhotometer® spectrophotometer (IMPLEN, München, Germany) and diluted with sterile double-distilled water to 50 ng/μL (Table 1).

### 2.2. Identification of Genome-Wide SSRs

MISA 2.1 software (Leibniz institute, IPK, Germany. http://pgrc.ipk-gatersleben.de/misa/ (accessed on 16 April 2021)) was used to search SSRs in the *H. molitrix* genome. SSR screening criteria were as follows: di-nucleotide repeats more than 6 times, tri-nucleotide repeats more than 5 times, tetra-nucleotide repeats more than 4 times, and penta- and hexa-nucleotide repeats more than 3 times. Compound SSRs were defined as the interval between two repeat motifs less than 100 bp. Due to the principle of complementary base pairing, the same kind of repetitive SSRs were merged as a repetitive representation. Di-nucleotide AC (AC/TG/CA/GT), tri-, tetra-, penta-, and hexa-nucleotides follow the same principles.

### 2.3. Primer Design for Genome-Wide SSRs

SSR primers were designed using the Primer 3.0 software according to the flanking sequence of SSRs. The design principles of primers were as follows: the primer sequence from the core sequence was 50~80 bases, the PCR amplification product was from 100 to 400 bp, and the annealing temperature was from 50 °C to 60 °C. GC content ranged from 40% to 60%.

### 2.4. Verification of SSRs Using PCR Amplification

Ninety-six pairs of SSR primers with three bases and above were designed and synthesized by Tianyi Huiyuan Biotech Company, Wuhan, China (Appendix A). The PCR reaction system contained 5.0 μL 2 × Taq PCR Master Mix, 1 μL template DNA (20 ng/μL), 0.5 μL of each primer (10 μL/mol), and DNase-/RNase-free deionized water 3.0 μL. Two-stage amplification programs were used. In the first stage, the pre-denaturation at 95 °C for 5 min caused the annealing temperature to gradually decrease from 62 °C to 52 °C, with a total of 10 cycles. The second stage included 25 amplification cycles, and the annealing temperature was 52 °C. In these two stages, the denaturation and extension steps remained unchanged for 30 s at 95 °C and 72 °C, respectively. After the second stage, the final extension was carried out at 72 °C for 20 min. Ninety-six pairs of primers were selected for PCR amplification and detected by 1% agarose gel electrophoresis. Finally, 13 SSR markers were obtained (Table 2). The PCR products were subjected to SSR analysis on an ABI 3730xl instrument, and then the genotype data were read using GeneMarker (Applied Biosystems).

### 2.5. Genetic Analysis 

POPGENE 1.32 [17] was used to calculate the number of alleles (Na), the number of effective alleles (Ne), expected heterozygosity (He), observed heterozygosity (Ho), Shannon’s index information (I), and Nei’s genetic distance. The polymorphism information content (PIC) was calculated by Cervus 3.0 software (Kruuk, Australian National University, Australian) [18]. The genetic differentiation index (Fst) of each population was calculated using Arlequin version 3.5 [19] and the molecular variance analysis (AMOVA) was performed.

The phylogenetic tree was constructed based on Nei’s genetic distance and an unweighted pair-group method with arithmetic mean using MEGA 5.0 [20]. Structure v2.3.4 [21] was used to evaluate the genetic relationship between populations. Based on the Bayesian model, the clustering value (K value) was found based on the hybrid model. The length of the burn-in period at the beginning of Markov Chain Monie Carfo (MCMC) was set to 50,000 times, and the range of K value was set to 1–8. Each K value was repeated 20 times. The analysis results were submitted to Structure Harvester (http://taylor0.biology.ucla.edu/struct harvest/ (accessed on 13 April 2022)) to determine the best K value, and then CLUMPP1.1.2 software (Rosenberg, Oxford University Press, USA) [22] was used for repeated clustering analysis. Finally, DISTRUCT1.1 [23] was used for visualization.

## 3. Results

### 3.1. Identification of SSRs in the H. molitrix Genome

A total of 368,572 SSR repeats were screened in the 842.01 Mb genome of *H. molitrix*. The total length of the identified SSRs was 6,492,076 bp, accounting for 0.77% of the total length of the whole genome. The average length of SSRs was 84.66 bp, the frequency was 437.73 loci/Mb, and the density was 7713.16 bp/Mb (Table 3). Di-nucleotide repeats (204873) accounted for 55.59% of the total number of SSRs, followed by tetr- (70,012,19%), pent- (44,921,12.19%), tri- (38,048,10.32%), and hexa-nucleotide repeats (10,718,2.90%). The highest frequency and density were Din- (243.31 loci/Mb, 5832.79 bp/Mb), followed by tetra- (83.15 loci/Mb, 846.65 bp/Mb), penta- (53.35 loci/Mb, 392.48 bp/Mb), and hexa-nucleotides (12.73 loci/Mb, 84.38 bp/Mb) (Table 3 and Appendix A).

Among different repeat types, AC (89,924) had the largest number of di-nucleotide repeats, accounting for 43.89%, followed by AT and AG, which were 40.6% and 15.4%, respectively. CG had the lowest proportion (0.1%). The highest frequency type of tri-nucleotide is AAT (24,409), accounting for 64.15%, followed by AAC (12.34%), AAG (5.36%), and the remaining repetitive sequences are relatively few. Among the tetra-, penta-, and hexa-nucleotide repeats, the highest number was AAAT (24%), AAAAT (28.2%), and AAAAAT (31.6%), respectively (Figure 1).

Among the di-nucleotide repeat types, AC (106.80 loci/Mb) had the highest distribution frequency, followed by AT (98.72 loci/Mb), AG (37.53 loci/Mb), and CG (0.27 loci/Mb). AT (3045.83 bp/Mb) had the highest density, followed by AC (2111.18 bp/Mb), AG (672.34 bp/Mb), and CG (3.44 bp/Mb). Among the tri-nucleotide repeat types, AAT (28.99 loci/Mb, 365.86 bp/Mb) had the highest distribution frequency and density, followed by AAC (5.58 loci/Mb, 63.67 bp/Mb), ATC (3.05 loci/Mb, 37.90 bp/Mb), and AAG (2.42 loci/Mb, 29.29 bp/Mb). Among the tetra-nucleotide repeat types, the distribution frequency of AAAT (19.97 loci/Mb) was the highest, followed by AGAT (17.52 loci/Mb), while the density of AAAT (178.24 bp/Mb) was lower than that of AGAT (208.83 bp/Mb). AAAAT (15.04 loci/Mb, 111.11 bp/Mb) and AAAAAT (4.02 loci/Mb, 24.32 bp/Mb) had the highest frequency and density in penta- and hexa-nucleotide repeat types (Table 4 and Appendix A).

### 3.2. The Distributions of Copy Numbers in Different SSR Repeat Types in H. molitrix Genome

In the *H. molitrix* genome, the copy number of di-nucleotide repeats ranged from 6 to 29 times, which accounted for 95.74% of the total di-nucleotide SSRs, and 6 repeat types were the most abundant (47,731) and accounted for 23.3%. The copy number of tri-nucleotide repeats was mainly concentrated in 5–10 times, accounting for 97.72% of the total number of tri-nucleotide SSRs, of which 5 repeats were the most (18,401), accounting for 48.36% of the total. The copy number of tetra-nucleotide repeats was largely concentrated in 4–8 times, accounting for 95.64% of the total number of tetra-nucleotide SSRs, of which 4 repeats were the most (35,997), accounting for 51.42% of the total number. The copy number of penta-nucleotide repeats was mostly concentrated in 3–5 times, accounting for 96.37% of the total number of penta-nucleotide SSRs, of which 3 repeats (33,442) accounted for 74.45%. The 3–5 times copy number of hexa-nucleotide repeats dominated, which accounted for 98.9% of the total hexa-nucleotide SSRs. Among them, the number of three repeats were the most (9683), accounting for 90.34% of the total number (Figure 2, Appendix A).

### 3.3. Distribution of SSRs on Chromosomes

The total length of 24 assembled chromosomes in *H. molitrix* accounted for 95.87% of the total assembled sequences. A total of 260,712 SSRs were screened, of which the largest number (15,637) was located in chromosome 1, accounting for 6%, followed by chromosome 2 (15,610) and chromosome 4 (14,548), accounting for 5.98% and 5.58%, respectively. The number of SSRs on chromosome 24 was the least (7650), accounting for 2.93% (Figure 3, Appendix A). Linear regression analysis was performed using SPSS, and the results showed that the total number of SSRs was positively correlated with chromosome length (R = 0.969, *p* < 0.01).

The frequencies of SSRs on 24 different chromosomes of *H. molitrix* were also different. The average frequency of SSRs was 323.22 loci/Mb. The highest frequency of SSRs was 347.31 loci/Mb on chromosome 4, followed by 343.67 loci/Mb on chromosome 10 and 342.63 loci/Mb on chromosome 23, and the lowest frequency was 276.45 loci/Mb on chromosome 11 (Figure 3, Appendix A).

The average density of SSRs was 12,499.87 bp/Mb. The highest density of SSRs was 15,035.65 bp/Mb on chromosome 24, followed by 14,456.46 bp/Mb on chromosome 23 and 13,499.65 bp/Mb on chromosome 12, and the lowest density was 9439.56 loci/Mb on chromosome 11 (Figure 3, Appendix A).

### 3.4. Screening of Polymorphic SSR Sites

A total of 56 alleles were detected in 13 SSR markers, observed number of alleles (Na) ranged from 2 to 7, effective numbers of alleles (Ne) ranged from 1.052 to 4.765, observed heterozygosity (Ho) ranged from 0.017 to 0.683, expected heterozygosity (He) ranged from 0.049 to 0.800, Shannon’s index information (I) ranged from 0.133 to 1.747, and polymorphism information content (PIC) ranged from 0.048 to 0.768 (Table 5). 

### 3.5. Population Genetic Diversity Analysis 

Na in four *H. molitrix* populations ranged from 3.538 (YW) to 4.462 (XC), with an average of 4.116; Ne ranged from 2.045 (YW) to 2.461 (WH), with an average of 2.307; Ho ranged from 0.392 to 0.450, with an average of 0.4138; He ranged from 0.402 (XC) to 0.504 (WH) with an average of 0.457, and the mean value of He was greater than that of Ho, indicating that the proportion of homozygotes was greater than that of heterozygotes. Shannon’s index information (I) ranged from 0.879 to 0.954, with an average of 0.911. The Fixation Index (Fst) in the group was between 0.084 (SS) and 0.178 (YW) (Table 6).

### 3.6. Genetic Differentiation in Four Populations

Analysis molecular of variance (AMOVA) was used to detect genetic differentiation among populations. The results showed that 95.35% of genetic variation was within populations, while only 4.65% was among populations. The Fst value among populations was 0.04650 (*p* < 0.001) (Table 7). Genetic differentiation index between YW and WH populations was the highest, while it was the lowest between XC and SS (Table 8). Based on Nei’s genetic distance, a population phylogenetic tree was constructed by the UPGMA method. The cluster results showed that SS, XC, and WH populations were clustered into one clade, and the YW population was clustered in another (Figure 4). A structure harvester was used to determine the best k value of 2, and it is inferred that the four populations can contain all individuals with the greatest possibility (Appendix A). Different colors in the clustering diagram represented different groups. The results showed that the YW population was significantly different from the SS, WH, and XC populations, and it was divided into two different groups (Figure 5). 

## 4. Discussion

In the *H. molitrix* genome, 368,572 SSRs were screened and accounted for 0.77% of the total genome length. The genome-wide SSRs content was similar to that of *Takifugu rubripes* (0.77%) and *Takifugu flavidus* (0.73%) [24]. It was higher than some lepidopteran insects (0.13–0.61%) [25] and birds (0.13–0.49%) [26], but lower than mammals such as *Homo sapiens* (3%) [27], *Bos mutus* (5.8%), *Bubalus bubalis* (5.69%) [28], and some macaque species (4.96–5.18%) [29]. These differences were caused due to studying species, genome size, and setting parameters for SSR screening.

The frequency of SSRs in *H. molitrix* was 437.73 loci/Mb, lower than that in *Cyprinus carpio* (621.95 loci/Mb), *Oncorhynchus kisutch* (461.35 loci/Mb), and *Cynoglossus semilaevis* (3445.94 loci/Mb) [30], and higher than that in some birds (80.9–256.9 loci/Mb) [26] and *Lateolabrax maculatus* (425.06 loci/Mb) [31]. The density (7713.16 bp/Mb) was higher than that of *Ctenopharyngodon idella* (1425.35 bp/Mb) [32] and lower than that of *Monopterus albus* (10,259 bp/Mb) [33]. 

Studies have shown that the longer the species evolution, the more SSR repeats of low repeat units there are. [34]. The dominant repeat type of SSRs in the *H. molitrix* genome is a di-nucleotide, which is consistent with that of most aquatic organisms, such as *Pelteobagrus fulvidraco* [35], *Hemibagrus wyckioides* [36], and *L. maculatus* [31]. This may be related to the evolutionary time of species.

Among the di-nucleotide repeat types, AC/GT had the largest number, which was consistent with most vertebrates [37]. The distribution of SSRs in different species exhibits certain differences, but the G/C bases in the genome are generally low [26]. AAT/ATT, AAAT/ATTT, AAAAT/ATTTT, and AAAAAT/ATTTTT are the dominant tri-, tera-, penta-, and hexa-nucleotide repeat types, respectively. A/T bases accounted for the majority of SSRs in the whole genome, while G/C content was less. This is similar to the results of *H. wyckioides* [36], *C. carpio* [38], and *Misgurnus anguillicaudatus* [39]. It was speculated that the CpG di-nucleotide sequence of cytosine (C) usually methylated, and then went through deamination to generate thymine (T) [40]. In addition, sequences containing A/T were prone to base sliding during replication, and G/C content was negatively correlated with the probability of replication sliding [41].

The number of SSR repeat copies in the *H. molitrix* genome was mainly concentrated between 3 and 29. The repeat number gradually decreased with the increase in the number of repeat unit copies, which was consistent with the distribution of SSRs in most genomes. The number of SSR repeats decreased with the increase in repeat length, because the longer the repeat length, the higher the possibility of mutation [42]. A large number of studies have shown that the number of SSRs on different biological chromosomes is correlated with their length. Linear analysis showed that the chromosome length of *H. molitrix* was positively correlated with the number of SSRs (R = 0.969, *p* < 0.01). The longer the chromosome, the higher the microsatellite content [43]. The frequency and density of SSRs was not correlated with chromosome length, which was relevant to the long-term evolution of species in 14 fish species [30].

Genetic diversity is an important genetic index to evaluate population adaptability, which can be estimated by the observed number of alleles, effective numbers of alleles, observed heterozygosity, and expected heterozygosity [44,45]. Many studies have shown that unintentional parental selection and inbreeding in the process of reproduction can lead to a decrease in the genetic diversity of populations. Our present results showed that the average values of Na, Ne, Ho, He, and I were 3.538–4.462, 2.045–2.461, 0.392–0.450, 0.402–0.5, and 0.879–0.954, respectively, which indicated that the genetic diversity among the four populations was low. This is consistent with [46] in Guangxi-cultivated *H. molitrix* populations. Therefore, the genetic diversity of the four cultivated *H. molitrix* populations in this experiment is low, which is very unfavorable to the protection of germplasm resources, and more scientific breeding measures should be carried out in the process of reproduction to improve the genetic diversity.

Fst is usually used to evaluate genetic differentiation among populations [47]. When 0 < FST < 0.05, there was no differentiation among populations, when 0.05 < FST < 0.15, there was moderate differentiation between groups, and when 0.25 < FST < 1, there was a high differentiation between groups [48]. Among these populations, the Fst values of the YW population and the other three populations (SS, WH, and XC) were 0.05402, 0.08709, and 0.06319, respectively. The YW population exhibited moderate differentiation from the other three populations. This is consistent with the results of the UPGMA tree and cluster diagram. 

## 5. Conclusions

In conclusion, this study analyzed the number, frequency, distribution, and type of SSRs in the whole genome of *H. molitrix*, screened and developed SSR markers at the level of the whole genome for the first time, and then analyzed the genetic diversity of four breeding populations. It was found that the genetic diversity of these four populations was low. Therefore, developing new SSR markers from the *H. molitrix* genome will provide a basis for genetic diversity analysis, the formulation of more scientific breeding measures, the protection and development of germplasm resources, and the realization and development of a sustainable aquaculture.

## Figures and Tables

**Figure 1 genes-13-01267-f001:**
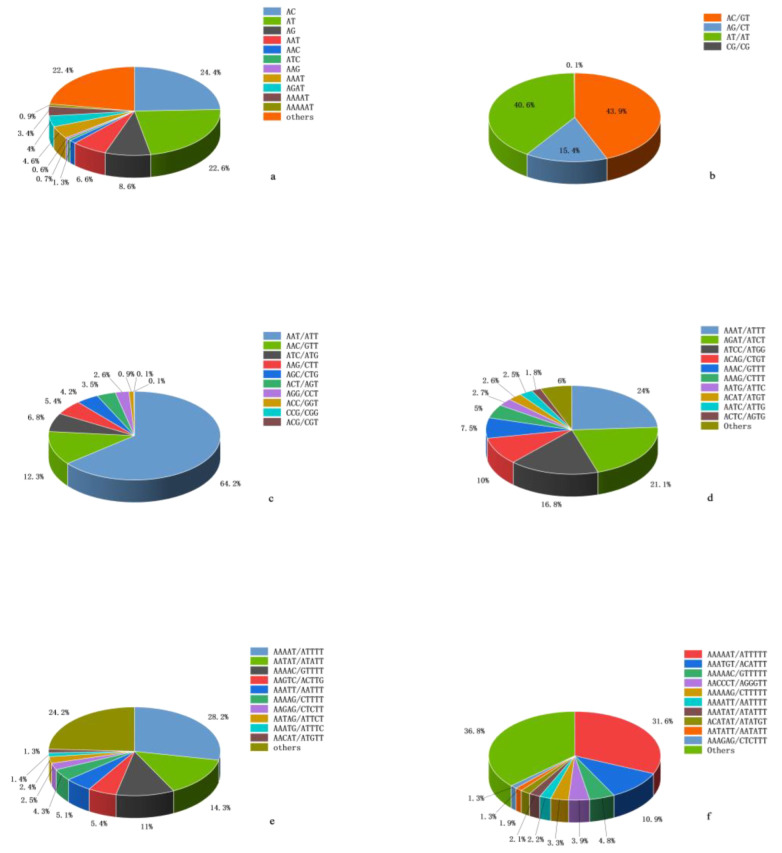
Motif proportions of different repeat types in *H. molitrix* genome. (**a**) The most abundant SSRs motifs in the *H. molitrix* genome; (**b**) mono-nucleotide repeat types, (**c**) tri-nucleotide repeat types, (**d**) tera-nucleotide repeat types, (**e**) penta-nucleotide repeat types, and (**f**) hexa-nucleotide repeat types.

**Figure 2 genes-13-01267-f002:**
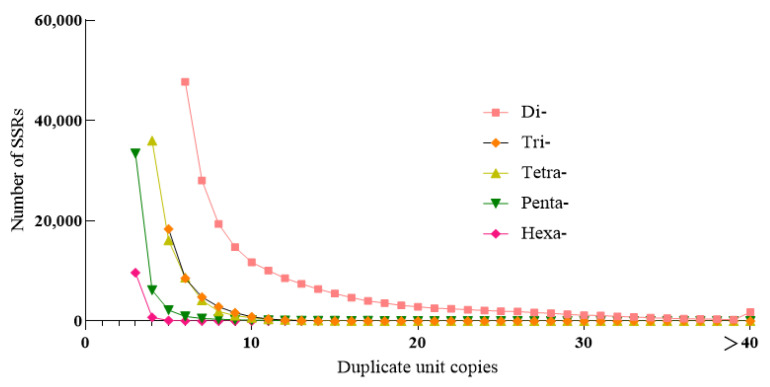
Different copy number distribution of *H. molitrix* SSRs.

**Figure 3 genes-13-01267-f003:**
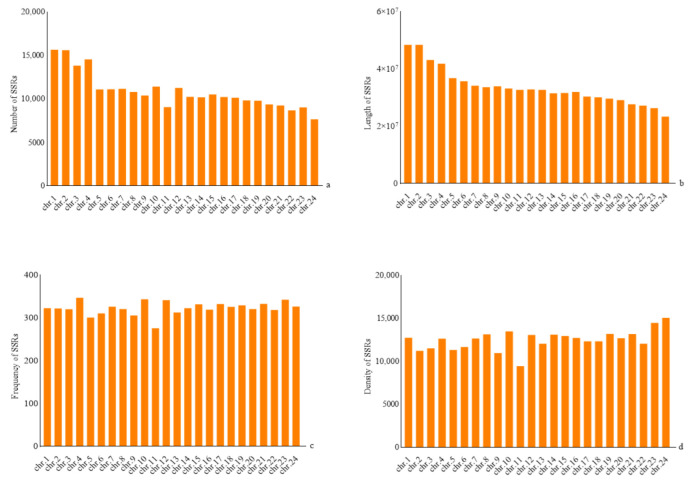
Analysis of SSR frequencies on *H. molitrix* chromosomes: (**a**) number of SSRs, (**b**) length of SSRs, (**c**) frequency of SSRs, and (**d**) density of SSRs.

**Figure 4 genes-13-01267-f004:**
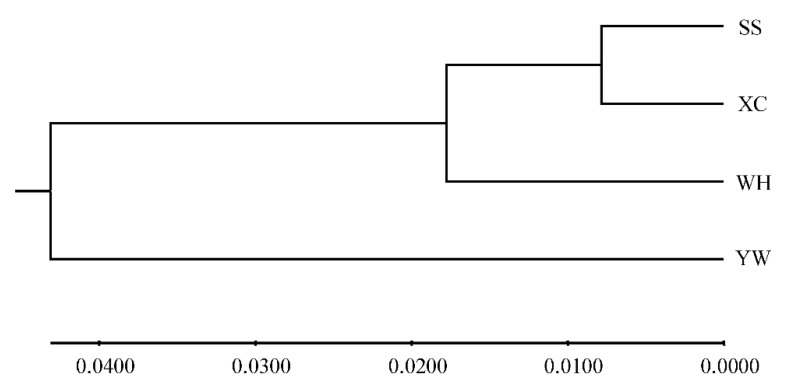
UPGMA tree based on the genetic distance among four *H. molitrix* populations.

**Figure 5 genes-13-01267-f005:**
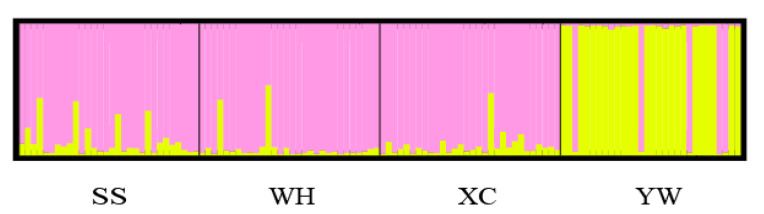
Structure analysis of *H. molitrix* populations using genotype data from 13 SSR sites. (K = 2).

**Table 1 genes-13-01267-t001:** Sample information of the cultured *H. molitrix* populations in this study.

Population	Location	Coordinates	Sampling Time	Number
SS	Shishou, Hubei Province, China	112°48′E,29°83′N	2021	30
WH	Wuhan, Hubei Province, China	114°48′E,30°77′N	2021	30
XC	Xiaochang, Hubei Province, China	113°59′E,30°76′N	2021	30
YW	Yaowan, Hubei Province, China	112°31′E,30°26′N	2021	30

**Table 2 genes-13-01267-t002:** Information of SSR markers analyzed in this study from *H. molitrix*.

Locus	Motif	Primer (5′–3′)	*T*_m_/°C	Size/Bp
P020	(AATA)_5_	F: CTGCTCACCTCAGCTCATCC R: ACATCACGGGAGCACAGAT	60	152~176
P021	(ATAG)_4_	F: GCGAGCGCATGTTTGATCAA R: GCGAGCGCATGTTTGATCAA	60	130~140
P030	(AAC)_5_	F: GGTATCTGCTCGCTGGATCC R: AATGCGCAGTTTCACAACG	60	201~213
P034	(ATA)_11_	F: GGGCGATGATCCCTGAATCC R: TGGGCGTTCTGGCACAATAT	60	199~221
P036	(ATT)_5_	F: GCTTGCTCAAGGGCACAATG R: TGCAGCAAGGACATTAGCGA	60	230~243
P037	(ATTAT)_4_	F: AGAGCACGTTCACCTCACTG R: CCGGCAATGCACAGTACAAG	60	230~273
P041	(TAG)_8_	F: AGAGGGAGACACGGCTACAT R: GAATGAGCGACCTCTAGCGG	60	212~240
P043	(ACA)_9_	F: TCACATCCTGCAACAGGGTC R: GTGTTCTGCCACCTTCCAGT	60	231~250
P054	(TAA)_6_	F: TTGTTCGCTCCTTGGAAGGT R: AAGATGGCTCAGGTTCACGG	60	241~278
P056	(TATC)_6_	F: CGACCTGCTAGCCCAAACAT R: GAAACGGAGACCTCTGGTGG	60	280~296
P058	(GTTT)_6_	F: AACTGTCTATGCGATGCCGT R: AATTTCATCCCGCAGTGCTG	60	280~293
P062	(TGTTT)_37_	F: ATGCTGGCGATATGTGGCAA R: ATACTCAGACCAGCCCGTCT	60	296~305
P086	(TAA)_5_	F: TATTGCAGTGGTCGGACACA R: ATACTGGGTTGCGCAGACTG	60	322~364

**Table 3 genes-13-01267-t003:** Information of SSR repeats in *H. molitrix*.

SSR Types	Total Counts	Total Length (bp)	Average Length (bp)	Frequency (loci/Mb)	Density (bp/Mb)	Percent (%)
Di-	204,873	4,911,266	23.97	243.31	5832.79	55.59
Tri-	38,048	468,878	12.32	45.19	556.86	10.32
Tetra-	70,012	712,888	10.18	83.15	846.65	19.00
Penta-	44,921	330,472	7.36	53.35	392.48	12.19
Hexa-	10,718	68,572	30.83	12.73	84.38	2.90
Total	368,572	6,492,076	84.66	437.73	7713.16	100.00

**Table 4 genes-13-01267-t004:** The most abundant motif categories in the *H. molitrix* genome SSRs.

Motif	Categories	Number	Frequency (loci/Mb)	Density (bp/Mb)	Length (bp)
Di-	AC	89,924	106.80	2111.18	1,777,632
	AT	83,122	98.72	3045.83	2,564,620
	AG	31,602	37.53	672.34	566,120
	CG	225	0.27	3.44	2894
Tri-	AAT	24,409	28.99	365.86	308,054
	AAC	4696	5.58	63.67	53,608
	ATC	2572	3.05	37.90	31,910
	AAG	2038	2.42	29.29	24,660
Tetra-	AAAT	16,819	19.97	178.24	150,084
	AGAT	14,755	17.52	208.83	175,838
Penta-	AAAAT	12,660	15.04	111.11	93,554
Hexa-	AAAAAT	3384	4.02	24.32	20,474

**Table 5 genes-13-01267-t005:** Characteristics of 13 polymorphic SSR sites in *H. molitrix*.

Locus	Na	Ne	Ho	He	I	PIC
P020	2	1.363	0.250	0.267	0.437	0.231
P021	2	1.342	0.267	0.255	0.423	0.222
P030	5	1.941	0.517	0.485	0.954	0.449
P034	3	2.044	0.450	0.511	0.819	0.442
P036	3	1.482	0.317	0.325	0.606	0.298
P037	3	1.052	0.017	0.049	0.133	0.048
P041	5	2.636	0.467	0.621	1.221	0.581
P043	7	3.084	0.600	0.676	1.446	0.643
P054	2	1.654	0.217	0.395	0.584	0.319
P056	7	4.332	0.683	0.769	1.639	0.735
P058	6	2.092	0.583	0.520	1.083	0.487
P062	4	2.332	0.283	0.571	1.047	0.520
P086	7	4.765	0.450	0.800	1.747	0.768
Mean	4.308	2.317	0.392	0.479	0.934	0.442
St. Dev	1.974	1.140	0.186	0.216	0.494	0.213

**Table 6 genes-13-01267-t006:** Statistical values of genetic diversity of 13 SSR sites in four *H. molitrix* populations.

Population	N	Na	Ne	Ho	He	I	Fst
SS	30	4.385 ± 0.583	2.387 ± 0.335	0.421 ± 0.071	0.469 ± 0.068	0.912 ± 0.520	0.084 ± 0.066
WH	30	4.077 ± 0.487	2.461 ± 0.330	0.450 ± 0.006	0.504 ± 0.063	0.954 ± 0.482	0.100 ± 0.068
XC	30	4.462 ± 0.489	2.334 ± 0.352	0.392 ± 0.075	0.451 ± 0.072	0.879 ± 0.525	0.103 ± 0.069
YW	30	3.538 ± 0.368	2.045 ± 0.386	0.392 ± 0.103	0.402 ± 0.067	0.900 ± 0.455	0.178 ± 0.132

**Table 7 genes-13-01267-t007:** Molecular variance (AMOVA) results of four *H. molitrix* populations.

Source of Variation	df	Sun of Squares	Variance Components	Percentage of Variation/%	Fixation Index
Among populations	3	29.704	0.12299	4.65	
Within populations	236	595.217	2.5221	95.35	Fst = 0.04650
Total variation	239	623.921	2.64509	100	

**Table 8 genes-13-01267-t008:** Genetic differentiation index (Fst) (above diagonal) and Nei’s genetic distance (below diagonal).

	SS	WH	XC	YW
SS		0.02718	0.00714	0.05402
WH	0.0328		0.03796	0.08709
XC	0.0157	0.0382		0.06319
YW	0.0673	0.1092	0.0823	

## Data Availability

The experimental data involved in this article can be obtained by the corresponding author.

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
