# Peer review of "Microsatellite Characteristics of Silver Carp (Hypophthalmichthysmolitrix) Genome and Genetic Diversity Analysis in Four Cultured Populations"

_genes, 2022, doi:10.3390/genes13071267_

Round 1
Reviewer 1 Report
The manuscript entitled "Microsatellite characteristics of silver carp (Hypophthalmichthys molitrix) genome and genetic diversity analysis in four cultured population", is of an interesting work. Authors have obtained several parameters to measure the diversity factors. Paper is written quite well and the topic could be interesting for the readers. I have some minor concerns-
1. After reading abstract, the message is not clear. Maybe author should modify the text by adding central message with concluding remark.
2. Why authors opted this study? What is the significance of doing genetic diversity analysis of Hypophthalmichthys molitrix?
3. Did authors try to apply Bayesian phylogenetic tress in this study to show the relationship among four cultured population?
4. author should add conclusion section in the manuscript after discussion.
Overall the manuscript is well written and the presentation is good.
Author Response
Dear Sir or Madam:
Thank you for your letter and the reviewers’ comments concerning our manuscript entitled “Microsatellite characteristics of silver carp (Hypophthalmichthys molitrix) genome and genetic diversity analysis in four cultured populations” (ID: 1797879). Those comments are all valuable and very helpful for revising and improving our paper, as well as the important guiding significance to our researches. We have studied comments carefully and have made correction which we hope meet with approval. Revised portion are marked in red in the paper. The main corrections and responds are as following:
Reviewer #1:
Point 1: After reading abstract, the message is not clear. Maybe author should modify the text by adding central message with concluding remark.
Response 1: Thank you. We have revised the section of Abstract according to your suggestion.
Point 2: Why authors opted this study? What is the significance of doing genetic diversity analysis of Hypophthalmichthys molitrix?)
Response 2: Hypophthalmichthys molitrix, as one of the four dominant fish in China, is widely distributed in ponds, lakes, rivers, and other major freshwater ecosystems in China and Asia. However, the germplasm resources of H. molitrix were degraded over the past few decades due to overfishing and environmental damage. Genetic diversity and structures are important genetic indexes to evaluate population adaptability, which is necessary to formulate more scientific breeding measures and protect and develop germplasm resources. Whereas more attention was paid to the analysis of genetic diversity and structure of natural populations before the implementation of the ten-year fishing ban in the Yangtze River. However, previously reports indicated that breeding populations might affect the genetic diversity and adaptability of natural populations through hatchery release of H. molitrix. In addition, there is no research on screening and developing SSR markers at the whole genome level to analyze the genetic diversity of H. molitrix. Therefore, the distribution characteristics of microsatellites in the assembled H. molitrix genome were analyzed, and new markers were developed to preliminarily evaluate the genetic diversity of the four breeding populations, which will provide basic data to analyze the degree of genetic variation and its temporal and spatial distribution in H. molitrix, and then take effective scientific means to protect genetic germplasm resources.
Point 3: Did authors try to apply Bayesian phylogenetic tress in this study to show the relationship among four cultured population?
Response 3: Thank you for your suggestion. Many previously reports showed that the UPGMA phylogenetic tree is reliable way and usually constructed based on genetic distance when microsatellite markers are used to study the genetic diversity among populations (Kariuki et al. 2021, Aquaculture; Zhou et al. 2021, Aquaculture Reports; Fang et al. 2021, Fisheries Research). Therefore, the UPGMA phylogenetic tree was applied to show the relationship among four culture population in our present study.
Point 4: Author should add conclusion section in the manuscript after discussion
Response 4: Thank you for your suggestion. We have joined the conclusion section at line 388 of the article.
We appreciate for your warm work earnestly, and hope that the correction will meet with approval. Once again, thank you very much for your comments and suggestions.

Reviewer 2 Report
This is a well-presented and well-executed piece of research, but I have a few concerns with the experimental design. I believe the work should be revised, and its purpose can be recast as a methodological analysis in accordance with my suggestions.
The authors sought to identify SSR characteristics of the silver carp (H. molitrix) genome that could be used to assess the genetic diversity of silver carp populations.
I have read this well-written and well-executed research paper, and while I am satisfied with the high quality of the laboratory analysis, I have a few reservations about the experimental design. The distinction between what the authors appear to assert and what they actually accomplished is thin but discernible.
The authors' use of samples from four populations of cultured stocks is not adequately explained or supported by the experimental design. For instance, why would one expect any cultivated stock from a specific region to exhibit a "model" of genetic diversity?
This is a limitation of the present work, in my opinion. Comparable to attempting to study the terrestrial diversity of life on the planet in a desert or the arctic. I do acknowledge the originality of the work, but only as a methodological advancement in the range of techniques used in ecological research or hatchery management for aquaculture. In this regard, the authors' arguments could be strengthened by referencing the work of Fang et al (2021, Fisheries Research, 235, 105829)
In addition, since the authors used POPGENE for the genetic analysis, they might also calculate the Shannon Index for each population to make a comparison with the other parameters examined in this study.
In a few places, the manuscript requires minor linguistic revisions and improvements.
-Lines 40-42. Please rephrase and expand, as it is unclear what you are criticizing in this sentence. I assume you are referring to the global phenomenon of efforts to restock fish populations and/or introduce new species. The work of Fang et al. (Fisheries Research, 235, 105829, 2021) may give support to your argument here.
Please rewrite lines 44-45 and correct the syntax
Author Response
Dear Sir or Madam:
Thank you for your letter and the reviewers’ comments concerning our manuscript entitled “Microsatellite characteristics of silver carp (Hypophthalmichthys molitrix) genome and genetic diversity analysis in four cultured populations” (ID: 1797879). Those comments are valuable and helpful for revising and improving our paper, as well as the important guiding significance to our researches. We have studied comments carefully and have made correction which we hope meet with approval. Revised portion are marked in red in the paper. The main changes in the paper and your related issues are presented below.
Reviewer #2:
Point 1: The authors sought to identify SSR characteristics of the Hypophthalmichthys molitrix genome that could be used to assess the genetic diversity of silver carp populations. I have read this well-written and well-executed research paper, and while I am satisfied with the high quality of the laboratory analysis, I have a few reservations about the experimental design. The distinction between what the authors appear to assert and what they actually accomplished is thin but discernible.
Response 1: Thank you very much for your rather positive evaluation and useful comments and suggestions on our paper. We explain and revise our manuscript as follows.
Point 2: The authors' use of samples from four populations of cultured stocks is not adequately explained or supported by the experimental design. For instance, why would one expect any cultivated stock from a specific region to exhibit a "model" of genetic diversity?
Response 2: Your suggestions are very important to us. The four Hypophthalmichthys molitrix breeding populations selected in this study were all over five years and used for reproduction, and 30 samples were collected. Therefore, it is representative to study the genetic diversity of these four aquaculture farms. Otherwise, the focus of this study is to screen and develop microsatellite markers at the genomic level of Hypophthalmichthys molitrix, and then preliminarily study the genetic diversity of four breeding populations. Next, we will continue to analyze more cultivated stock so as to obtaining more comprehensive genetic diversity of Hypophthalmichthys molitrix.
Point 3: The authors' arguments could be strengthened by referencing the work of Fang et al (2021, Fisheries Research, 235, 105829)
Response 3: Thank you very much for your valuable advice. We reinforce our views in the introduction based on research by Fang et al. (2021, Fisheries Research, 235, 105829).
Point 4: In addition, since the authors used POPGENE for the genetic analysis, they might also calculate the Shannon Index for each population to make a comparison with the other parameters examined in this study.
Response 4: According to your suggestion, we add the Shannon Index (I) to our manuscript, together with the alleles (Na), effective alleles (Ne), and observed heterozygosity (Ho), to evaluate the genetic diversity of Hypophthalmichthys molitrix.
Point 5: In a few places, the manuscript requires minor linguistic revisions and improvements.
Response 5: We revised our manuscript carefully and made all the necessary editorial changes. With all the changes we have made, we wish that this manuscript has been clarified and strengthened.
Point 6:Lines 40-42. Please rephrase and expand, as it is unclear what you are criticizing in this sentence. I assume you are referring to the global phenomenon of efforts to restock fish populations and/or introduce new species. The work of Fang et al. (Fisheries Research, 235, 105829, 2021) may give support to your argument here.
Response 6: Your proposal is quite good. We have referred the research of Fang et al. (Fisheries Research, 235, 105829, 2021) to support our argument.
Point 7:Please rewrite lines 44-45 and correct the syntax
Response 7: We have re-written this part according to your suggestion.
We appreciate for your warm work earnestly, and hope that the correction will meet with approval. Once again, thank you very much for your comments and suggestions.

Round 2
Reviewer 2 Report
Your work was revised to satisfactory